# Research Progress of MEMS Inertial Switches

**DOI:** 10.3390/mi13030359

**Published:** 2022-02-24

**Authors:** Min Liu, Xinyang Wu, Yanxu Niu, Haotian Yang, Yingmin Zhu, Weidong Wang

**Affiliations:** School of Mechano-Electronic Engineering, Xidian University, Xi’an 710071, China; mliu_12@stu.xidian.edu.cn (M.L.); wuxinyang_1999@163.com (X.W.); nikx_1999@163.com (Y.N.); 17040210008@stu.xidian.edu.cn (H.Y.); ymzhu@xidian.edu.cn (Y.Z.)

**Keywords:** MEMS, inertial switch, threshold acceleration, sensitive direction, contact effect, fabrication

## Abstract

As a typical type of MEMS acceleration sensor, the inertial switch can alter its on-off state while the environmental accelerations satisfy threshold value. An exhaustive summary of the design concept, performance aspects, and fabrication methods of the micro electromechanical system (MEMS) inertial switch is provided. Different MEMS inertial switch studies were reviewed that emphasized acceleration directional and threshold sensitivity, contact characteristics, and their superiorities and disadvantages. Furthermore, the specific fabrication methods offer an applicability reference for the preparation process for the designed inertial switch, including non-silicon surface micromachining technology, standard silicon micromachining technology, and the special fabrication method for the liquid inertial switch. At the end, the main conclusions of the current challenges and prospects about MEMS inertial switches are drawn to assist with the development of research in the field of future engineering applications.

## 1. Introduction

Inertial switches are more and more widely used in automotive electronics, inertial navigation, and weapon equipment products [1,2,3,4,5,6]. They can be nested in wearable systems, transport devices for monitoring health, maintenance needs, and transportation of special goods [2,7,8,9]. As the rapid expansion of the Internet of Things (IoT) [10] and the requirements of the application environment increase, the research interest in MEMS inertial switches is growing [9,11,12]. Compared to piezoelectric microswitches, electrostatic microswitches, and accelerometers, inertial MEMS switches have some distinctive characteristics, such as less power consumption, a simple structure, and an interface circuit, which can maintain strong durability in environments with inconvenient power supply. However, accelerometers always consume a certain amount of power even without an external excitation [13,14,15]. The main disadvantages of electrostatic and piezoelectric microswitches are their high drive voltage and low reliability [16,17,18]. An inertial MEMS switch combines sensing and actuation. When sufficient acceleration is sensed, the movable electrode contacts the fixed electrode and the external circuit is triggered [19]. This kind of switch with only acceleration excitation is also superior to electrostatic and piezoelectric accelerometers because they avoid electromagnetic interference in applications [3].

Based on the state of the inertial switch “on” after the acceleration excitation is removed, inertial microswitches have been classified into two categories: the intermittent switch and the persistent switch. The intermittent switch refers to the switch resuming the initial disconnected condition after the acceleration dissipates, and the persistent switch means that the switch performs a keep-closed function after the acceleration is evacuated. The most obvious feature of the intermittent switch is that the switch does not have self-blocking capability. Intermittent inertial switches work on the principle that the movable electrode (mainly a suspended mass) comes into contact with a fixed electrode when the acceleration reaches its threshold. The model of a spring–mass system is shown in Figure 1 [1,11,20]. Here, the intermittent inertial switches are mainly classified and discussed by means of the different properties containing the following three groups: directional switches, various acceleration threshold switches, and contact enhanced switches. For the persistent inertial switch, the realization of the locking function is carried out by various structures, so the classification is mainly dependent on the feature structures. These switches are embodied in three aspects, including latching switches [8,21,22], bistable switches [23,24,25,26], and microfluidic switches [27,28,29,30,31]. Persistent switches have an excellent contact effect due to special structural features or external auxiliary units.

## 2. Intermittent Inertial Switches

### 2.1. Sensitive Direction

Axial sensitivity has always been an important research topic for MEMS inertial devices. In the actual working environment, the inertial switch is inevitably impacted in different directions. For the inertial switch, it is important to prevent spurious triggering from shocking interference in non-sensitive directions. Therefore, reducing the disturbances and improving the sensitivity of the working axis of the device have attracted the attention of many researchers. MEMS inertial switch studies with different direction sensitivity are summarized in the following sections.

#### 2.1.1. Uniaxial Inertial Switches

A uniaxial sensitivity inertia switch has only one sensing direction. The main structure of a uniaxial inertial switch is a well-known spring–mass damper system. Consider the vibration model of a traditional single-degree-of-freedom switch system (Figure 1) under acceleration (*a*) excitation. The dynamic equilibrium model of the MEMS inertial switch can be described with Equation (1).
(1)$$m\ddot{y}+c\dot{y}+ky=ma$$
where *m*, *y*, *c*, and *k* represent the weight and the displacement of the proof mass, the squeeze film damping coefficient, and the stiffness of the spring beam, respectively.

When the inertial switch senses an acceleration (i.e., equal to or at more than the threshold level) in the sensitive direction, the movable electrode overcomes the damping force and moves along the sensitive direction. When the movement displacement is greater than the initial gap *y*_0_, the movable electrode contacts the fixed electrode and the inertial switch is turned on. When the acceleration signal is removed, the movable electrode is pulled back under the action of elastic force, which finally returns to its initial position.

At present, a variety of single-axis inertial switches have been reported. For a uniaxial sensitive inertial switch, it is necessary to increase the structural stiffness in the insensitive directions, as then the device is less susceptible to disturbance from off-axis or rotational accelerations. In order to obtain a lower off-axis sensitivity inertial switch, the key method is designing a symmetrical structure for the proof mass, or using the constrained structure to limit the off-axis movement of the action unit.

Yang et al. [32] designed a vertically driven MEMS inertial switch. The switch consisted of the proof mass as the movable electrode and two compliant bridge beams as the fixed electrode. The proof mass was suspended by two conjoined serpentine springs. However, the first two frequencies of this kind of device by modal analysis are discussed in the subsequent study [33]. It was seen that the first and the second frequencies were very close, which indicates that the cross-coupling effect of the microswitch was very obvious. They improved the previous design of the fixed electrode in order to prolong the contact time of the switch. A new bridge-type fixed electrode was applied, but this design limited where the suspension springs of the mass could be placed, resulting in the suspension springs only being arranged on both sides of the head and tail of the mass. This design of the fixed electrode made the switch affect the off-axis accelerations easily. Therefore, they redesigned the switch, designed the device structure as a centrally symmetric structure, and changed the fixed electrode from two bridge beams to a cross beam to reduce off-axis sensitivity [34], as well as improve the fixed electrode overload capability. The ratio of the second-order and first-order frequencies of the simulation increased from 1.04 to 5.21, indicating that the off-axis sensitivity of the redesigned device was largely reduced. The threshold acceleration of the redesigned MEMS inertial switch was 70 g and the tested turn-on response time was 30 μs.

Wang et al. [7] designed a novel horizontally driven inertia microswitch with an elastic cantilever as the fixed electrode. By minimizing the clearance, the reverse-resist block worked well to reduced rebound force and enhance the contact time to 230 μs. The unique design of the sensitive mass improved the overall stability and reliability of the switch. Zhang et al. [35] proposed a laterally-driven MEMS inertial switch. The multi-directional tightly constrained structural design effectively reduced off-axis sensitivity and improved impact resistance. The constrainting plate with holes above the sensitive mass limited the radom motion of the proof mass and reduced the off-axis sensitivity of the inertial microswitch. In order to avoid rotation, in the new design of the inertia switch a symmetrically distributed double-layer serpentine spring beam was introduced [36]. When the inertial switch was impacted in the sensitive direction, there was no collision between the sensitive mass and the constraining structure, which improved the uniaxial sensitivity.

As mentioned above, many uniaxial inertial switches are constructed from a proof mass suspended by springs. Some schematic illustrations of these are displayed in Figure 2. Their vibration models are described in Equation (1). When these systems act as shock switches, according to the S.R.S., for the pulse time of acceleration, reliability is a serious problem. These traditional single-mass inertial switches may misoperate at shock levels less than the threshold, resulting in reduced switch reliability. Fathalilou et al. [37] proposed a dual-mass MEMS shock switch. An auxiliary mass spring was attached to the main mechanism of the switch, and the safety performance of the sensor under shocks below the threshold was improved by adjusting the auxiliary system. In addition, Ren et al. [38] proposed a self-powered MEMS inertial switch for a potential wake-up application without additional power consumption. When the switching device sensed a small vibration acceleration, the suspended mass vibrated continuously at the equilibrium position. During the movement of the movable mass, the capacitance of the parallel plates also changed, causing induced charge transfer. At this time, the alternating current of the device was converted into direct current, which could be stored in an energy storage device using a self-charging unit. Once an exceeding threshold acceleration was sensed, the movable electrode contacts the fixed electrode and carried out the circuit conduction. Then the charges stored in the energy storage device were released. When this device was impacted by an acceleration of 40 g, it generated a pulse signal.

Switches used in extreme acceleration environments (such as ballistic rockets) need to withstand accelerations of up to 100,000 g and operate with high fidelity at a low impact of 100 to 10,000 g. Raghunathan et al. [39] presented a novel MEMS low-g switch with serpentine flexures based on single-crystal silicon, which could withstand high load shocks 200 times the trigger load. The switch was capable of responding at low acceleration (60–131 g) and withstanding a high-g impact shock acceleration load of 24,000 g.

**Figure 2 micromachines-13-00359-f002:**
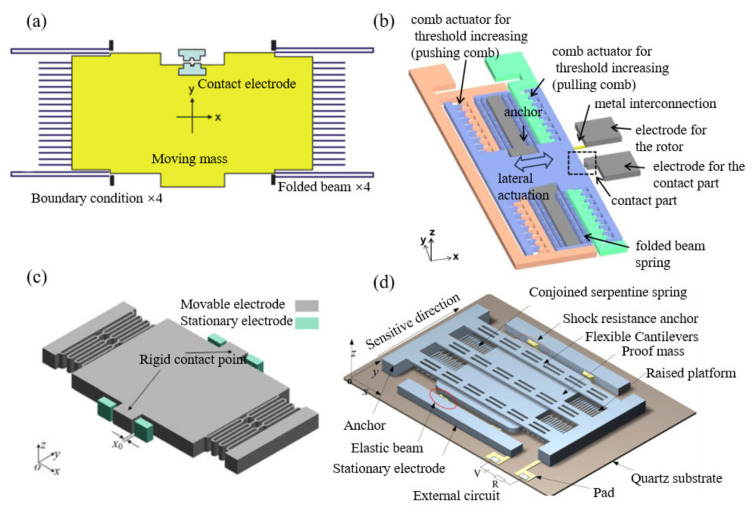
Some uniaxial inertial switches: (**a**) [40], (**b**) [12], (**c**) [41], (**d**) [42].

##### 2.1.2. Biaxial Inertial Switches

An inertial switch with biaxial sensitivity means that it can operate in two axes. There are two main methods to achieve biaxial directional sensitivity for the switch. One of the ways is to assemble two single-axial switches into one chip, and the new switch can realize biaxial sensitivity in a plane [43]. Although the direct assembly of single-axis switches improves the directional sensitivity, it also has some problems and defects, including assembly errors [44], excessive bulk [45], packaging difficulty [46], and high manufacturing costs [45]. Another method is to utilize multiple springs to support the mass, and the fixed electrode is placed around the mass in multiple directions to achieve biaxial sensitivity. That layout allows a more uniform distribution of the stiffness of the mass–spring system in all directions. Greywall et al. [47] claimed patents for two biaxial inertial switch designs. Both designs had a circular proof mass suspended by springs, but varied in the fact that the fixed electrodes were outer and inner. Lin et al. [48] presented another design of a two-axis low-g acceleration threshold switch. The switch utilized buffer springs connected to the proof mass and contacted the fixed electrode, which would reduce the bounce of the switch when the shock exceeded the acceleration threshold, thereby prolonging the contact time of the switch, as illustrated in Figure 3. Niyazi et al. [49] designed a novel bidirectional multi-threshold mems inertia switch capable of acceleration-based actuation of separate circuits. The design implemented a spring-suspended mass with fixed electrodes positioned in the sensitive direction, consuming no power in the stationary state. The acceleration thresholds were 69 g and 121 g. Xu et al. [50] fabricated a dual-axis MEMS shock sensor that could be triggered in each direction in a single plane. The device consisted of a movable electrode, which was a proof mass suspended by four serpentine springs, and three cantilever beams forming the first, second, and third stationary electrodes. The switch could sense an acceleration range of 800–2600 g.

##### 2.1.3. Triaxial Inertial Switches

A triaxial inertial switch can sense an acceleration signal in the *x*, *y*, and *z* axes, which usually includes the proof mass and three pairs of fixed electrodes in the orthogonal sensitive directions. Chen et al. [2] introduced an all-metal contact-enhanced triaxial inertial microswitch with low axial disturbance, which consisted of a proof mass as the movable electrode, and the fixed electrode of the switch in the horizontal and vertical directions composed of two L-shaped flexible cantilever beams and a multi-hole crossbeam. The acceleration threshold in the horizontal direction was 255 g and the contact time was 60 μs, whereas in the vertical direction, the threshold acceleration was 75 g with a contact time of 80 μs. Currano et al. [51] designed a triaxial MEMS inertial switch using a single mass–spring assembly. The device consisted of an annular mass and a centrally clamped suspension composed of four spiral springs. The spiral spring was used to realize compliance in all directions. The switch had an acceleration threshold range between 50 g and 250 g.

##### 2.1.4. Multidirectional/Omnidirectional Inertial Switches

Omni-directional switches can sense a shock signal in all the radial directions of a hemisphere. For the actual engineering environment, the acceleration excitation may come from any direction. If a single-axis or dual-axis switch is simply combined to monitor multi-directional acceleration, it is easy to cause installation errors and centroid deviations, affecting the reliability of test results. Besides, the fabrication cost of the switch increases. Therefore, it is of great practical significance to develop triaxial or omnidirectional inertia switches. Such switches have been widely applied in the field of military weapons. Inertia switches are often used in automotive airbags and vibration monitoring fields.

Xi et al. [19] presented a novel MEMS omnidirectional inertial switch consisting of a proof mass and an axial and four radial flexible electrodes to form a dual mass–spring system. The switch had omnidirectional sensitivities in a half sphere. The switch could sense an acceleration range from 380 g to 500 g. Liu et al. [52] presented a MEMS-based low-g inertial switch that consisted of an annular proof mass suspended by a helix spring. The helix spring could sense a weak signal. The acceleration threshold was 20 g. Yang et al. [53] designed a multidirectional-sensitive inertial microswitch with an electrophoretic flexible composite fixed electrode to enhance the contact effect. The contact time of the switch was prolonged because the new vertical composite fixed electrode was capable of reducing the stiffness, and the threshold was around 70 g, which was uniform in different sensitive directions. The fully axisymmetric serpentine spring was used to support the sensitive mass to sense any radial acceleration in the *xoy* plane. The T-shaped cantilever with a maple leaf shape on the top was used as a vertical fixed electrode, which could monitor the acceleration of the *z*-axis. Du et al. [54] studied a low-g omnidirectional microswitch via non-silicon surface micromachining technology, and the distribution of the threshold acceleration was uniform. The device consisted of combination of an annular-shaped proof mass and supporting Archimedes spiral springs as the moving electrode and static electrode located in the center of the ring. The structural sketch of the proposed inertial switch is shown in Figure 4. The threshold acceleration of the switch dropped to 38 g. Moreover, they improved the design, redesigning the structure of the central springs [55]. The switch adopted an S-type flexible connection that could reduce the stiffness of the electrode to extend the contact time of the switch. The dynamic threshold of the switch was 7.9 g to 11.3 g, and the response time in each direction was less than 2 ms. Chen et al. [56] designed an omnidirectional inertial switch with a rectangular spring. The omnidirectional switching system’s stability was reinforced, due to the design of rectangular springs. The threshold acceleration of the designed inertial switch was about 58 g in the x direction and 37 g in the z direction, and the contact time was about 18 µs. A summary of some examples of typical sensitive direction inertial switch performance is shown in Table 1.

### 2.2. Threshold Acceleration

#### 2.2.1. Low-g Threshold Inertial Switches

MEMS inertial switches sensing a low-g threshold acceleration can find representative application in ISDs (ignition safety devices), consumer electronics, and automotive and aerospace domains.

Chen et al. [57] designed a nickel low-g MEMS inertial switch with horizontal sensitivity, as shown in Figure 5a. The forward novel L-shaped elastic cantilever beams were used as the fixed electrode. The threshold acceleration was about 18 g, with the contact time of the switch around 216 μs. However, switches made of metal are not suitable for missile equipment with operating temperatures between −51 °C and 75 °C. Deformation due to thermal expansion of the metal would significantly change the acceleration threshold. When the inertia switch is used in the aerospace field, mainly in the climb and descent stages of an airplane, the requirement of the acceleration threshold is less than 30 g. Xiong et al. [58] designed a silicon-based low-g MEMS inertial switch with an acceleration threshold of 7.4 g, as shown in Figure 5b. The structure of the device was mainly composed of a large-volume mass block and a low-stiffness coil spring. Rectangular springs were designed in a spiral shape with a thin thickness in order to reduce the stiffness of the switch and facilitate flexible movement of the mass under applied load. Moreover, to further decrease the threshold, they redesigned the structure of the spring–mass system [59], as shown in Figure 5c. By reducing the spring thickness and increasing the spring length, the design obtained lower stiffness and the threshold value was 5 g. Furthermore, they introduced the conceptual design of a new MEMS inertial microswitch, which adopted the method of direct contact sensing to eliminate the bouncing effect when the electrodes were in contact. The measured threshold was approximately 11.8 g [60]. Field et al. [61] studied a low-g switch with a threshold value of 25 g. The fabrication was based on a silicon-on-insulator (SOI) wafer, and the sensing direction was along the thickness of the wafer. Hwang et al. [62] designed a low-g MEMS acceleration switch with a threshold acceleration of 6.61 g. The structural material was single-crystal silicon, the structure was stress-free, and the stability was good at high temperature. However, common SOI wafers have an unexpected disadvantage in that the balance of structure is hard to maintain, mainly due to over-etching by poor etching verticality, which causes quality deviation, shifts in the center of mass and the contact area becoming smaller, and the mass tilting or rotating when it moves along the shock direction, potentially leading to contact uncertainties. One solution is to adopt a double-buried layer SOI wafer to replace the original common SOI wafer. Massad et al. [63] designed a low-g inertial switch made of silicon, whose plate was suspended by four folded beams that acted as springs. The electrodes closed when an acceleration from 6 g to 10 g was applied.

#### 2.2.2. High-g Threshold Inertial Switches

In recent years, owing to the intelligent requirements of weapon systems, MEMS technology has also been closely integrated with advanced weapon systems. In smart ammunition fuse applications, the survivability of MEMS inertial switches under high mechanical loads is a key to measure the performance of switch devices. More miniaturization, strong anti-interference ability, high sensitivity and reliability of high-g MEMS inertial switches have been studied to better adapt to the development trend of military weapons.

Nie et al. [64] presented a MEMS high-g electronic inertial switch for medium- and large-caliber projectile fuses. This switch mainly consisted of four parts: a zigzag groove in the center of the sensitive mass, a latch, an electrical connection structure, and a movement-limiting mechanism, as shown in Figure 6. The zigzag groove had a damping effect on the mass movement, which had the function of sensitively distinguishing the fuse launch acceleration and accidental fall shock. The switch could withstand accelerations of 3000 g. A method for identifying the acceleration load of the zigzag groove of the mass block was established, which was of great significance for the parameter optimization of the switch structure under different accelerations. Singh et al. [65] designed a passive high-g inertial switch for critical applications without electricity. The switch consisted of a serpentine spring–mass system, the dimensions of which were optimized for the natural frequency [66], and the switch was closed at 3500 g acceleration. Xu et al. [67] presented an inertial microswitch with synchronously following flexible electrodes, and its spring-shaped structure was used not only to prolong the contact duration but also to reduce shock bounce. The switch achieved a contact time of 350 us at 500 g acceleration. Xi et al. [68] proposed a high-g inertial switch consisting of five individual electrical switches and a sensitive mass–spring system. This switch could detect not only the acceleration threshold, but also the direction of acceleration in three-dimensional space. The contact time of the switch was 119 μs when the acceleration shock was 1200 g.

#### 2.2.3. Threshold-Tuning Inertial Switches

If the threshold acceleration of the inertia switch is adjustable, its application can be extended to a wider range of environments. Kim et al. [12] designed a bidirectional adjustable acceleration switch that could simultaneously increase and decrease the threshold acceleration in the same mechanism. In their previous paper [69], they proposed a tunable acceleration switch, but the switch design could not decrease the threshold acceleration and the tunability was limited. It could be used in secure/armed position switching and mechanical locking adjustable switches for military applications. Threshold acceleration could be increased or decreased by electrostatic forces created by the electrical potential between the contact portion and the comb structure. The threshold acceleration without tuning voltage was 10.25 g, and then it was tuned to 2.0 g and 17.25 g by applying 30 V to the push and pull combs, respectively. Subsequently, a fully digital MEMS accelerometer was implemented using the concept of a MEMS accelerometer switch [70]. The accelerometer consisted of a proof mass and a number of parallel plate electrostatic actuators that could be turned on and off in a sequential manner by a digital controller (as shown in Figure 7). By changing the bias voltage and working voltage of the electrode, the full-scale acceleration and acceleration range of the most significant bit and least significant bit could be set to any value, thereby adjusting the acceleration range of the device. This design realized the adjustment of threshold acceleration between 0 and ~1 g. Ma et al. [71] presented an inertial switch with adjustable acceleration thresholds based on a MEMS digital-to-analog converter (M-DAC). The selected adjustable plates were pushed by diverse PDMS caps to produce a specific displacement of the sensitive mass, thus realizing the adjustment of the acceleration threshold. By using different upper PDMS caps, the acceleration thresholds changed between 40 g and 75 g. Abbasalipour et al. [72] expanded on the original design of the fully digital MEMS accelerometer and designed a micromechanical accelerometer with 8-bit digital control. The tuning force of each electrode group was three times that of the adjacent electrode group. The full-scale acceleration of the tested 8-bit prototype was 2.7 g. Table 2 shows a summary of some examples of typical threshold acceleration inertial switch performance.

### 2.3. Contact-Enhanced Inertial Switches

Solid–solid rigid contact can easily cause contact bounce, making it unsuitable for small signal switches. It also causes arcing and welding of the contacts, resulting in contact surface damage and material transfer. This contact degradation restricts the reliability and longevity of these devices. Avoiding solid-to-solid mechanical contact means there are no issues associated with contact bounce or contact degradation, thus extending the life of the switch.

Contact time is a critical property for an intermittent inertial switch in some application environments because the longer contact time, the less difficult the signal processing is. A short contact time requires an external circuit with higher signal identification performance [3]. Insufficient contact between two electrodes is usually caused by a rigid contact process, resulting in a strong rebound of the movable electrode after touching the fixed electrode (usually a rigid substrate for conventional inertial switches). These switches have extremely short contact times (less than 10 μs), which is difficult for the external circuit to recognize [1]. Furthermore, contact bounce can damage the interface between the two motors due to mechanical hammering and arcing, subsequently affecting the durability of the system, which may result in permanent adhesion between the two electrodes [73]. In recent years, in order to eliminate the bounce phenomenon and enhance the contact effect of the intermittent switches, considerable efforts have been put into inertial microswitches. The methods mainly include some special structures and flexible materials.

#### 2.3.1. Special Structures to Extend Contact Time

Cai et al. [1] developed a new double mass–spring system that is completely different from the traditional system design. In addition to the proof mass–spring system, instead of being rigidly fixed to the proof mass, the contact point was suspended in the middle of the proof mass by an inner spring. The experiment results demonstrate that the threshold acceleration of the MEMS inertial switch was about 175 g and the contact time of the contact point was 56 μs. The steady turn-on time of more than 50 μs was much longer than the 12 μs in the research [74]. The switch was driven vertically and the elastic beam above the proof mass block could be used as a fixed switch electrode. However, there were two switch-on signals under an acceleration of 200 g acceleration, which would lead to secondary closure of the switch and have a bad effect on the reliability of the switch. Wang et al. [5] proposed a new type of L-shaped flexible cantilever fixed electrode, and long contact was possible in a transverse MEMS inertial switch. The sensitive mass was suspended by two pairs of conjoined serpentine springs as the motion electrode and the fixed electrodes were suspended from the blocks. The flexible cantilever electrode could realize a flexible deformation to enhance contact time and avoid the rebound phenomenon of the MEMS inertial switch. The contact time was extended to 1050 μs, which was much longer than the time without cantilever buffer effect of 5 μs. Moreover, according to the characteristics of the cantilever arm, its special elastic deformation could be designed to realize the control method of switch contact time. With the increasing acceleration, the contact time also decreased until it tended to a stable value. At the same time, based on the same inertial switch structure, Chen et al. [3] analyzed the influence of the applied acceleration loads on contact time. Combining the tested results, it was concluded that as the pulse width of the applied shock load increased, so did the contact time. Yang et al. [32] investigated an inertial microswitch with bridge-type elastic fixed electrodes for prolonged contact. Specifically, they designed two parallel perforated elastic beams as fixed electrodes. The microswitch was equipped with a relatively good contact effect, but the switch could be triggered with an effective contact of only 12 μs when 100 g acceleration was applied. By adding a group of cantilever beams on the mass block as the buffer between the electrodes [73], the contact time could be prolonged and the phenomenon of jumping contact could be avoided. Another advantage of this is that the contact time of the microswitch could be extended to 240 μs. It was more than 15 times that of a microswitch without cantilever beam. Table 3 shows a comparison of contact times for simulated and experimental tests of three kinds of inertial switches. In addition, Yang et al. [33] also discussed three other devices with different structures (types II, III, and IV), as shown in Figure 8. Types II and III used a cross-beam stationary electrode to realize a long contact time with deformation. The contact times for the two switches (types II and III) utilizing beam deformation were 20 μs and 30 μs, respectively. However, due to the limitation of deformation, the contact effect of the switch was not effectively improved. Therefore, type IV adopted a double spring–mass system, which greatly improved the contact effect. The experimental results show that the contact time was about 55 μs, which greatly enhanced contact performance. Xu et al. [6] introduced a laterally actuated inertial switch. As a moving electrode, the sensitive mass was connected to two L-shaped elastic cantilever beams. The switch had an inductive acceleration threshold of 288 g with a contact time of 150 μs.

#### 2.3.2. Materials and Assistive Force to Extend Contact Time

Unlike the inertial switches described above with specific structures to increase contact time, some studies have proposed the use of special materials or external assistive forces to improve contact characteristics. Lee et al. [75] proposed a new inertial microswitch. Its contact mode was carbon nanotubes (CNTs) in contact with carbon nanotubes. Due to the high mechanical flexibility and elasticity of CNT material itself, it was prone to elastic deformation when used as contact pad material [76], so the contact time was prolonged. At present, the contact time of the existing traditional switch is only 7.5 μs, but with this inertial micro switch, the contact time could be extended to 114 μs. The results show that the new inertial microswitch proposed by Lee et al. could significantly prolong the contact time. Yang et al. [77] developed an inertial microswitch with polymer metal composite as a fixed electrode. It not only had the characteristics of multidirectional features and high sensitivity, but also prolonged the contact time of the switch and improved the stability and reliability of the inertial microswitch. The contact time was improved to 110 µs, which was longer than that one without polymer (~65 µs). In addition, an inertial microswitch with a flexible carbon nanotube/copper (CNT/Cu) composite array layer between movable and fixed electrodes was designed in [78]. The interaction between carbon nanotubes, i.e., adhesion, was one of the main reasons for the extension of contact time. Li et al. [79] designed a MEMS inertial switch to prolong the contact time through electrostatic assistance and multi-step pull action. Through the test, the contact time was about 540 μs and showed no bounce. Table 4 shows a summary of some examples of typical contact-enhanced inertial switch performance.

## 3. Persistent Inertial Switches

Compared to an intermittent inertial switch without a self-locking function, a persistent inertial switch can achieve a stable contact property with a mechanical lock and a bistable structure. Furthermore, a fluidic MEMS inertial switch can also accomplish a locking function when flowing liquid (as a movable electrode) contacts the metal fixed electrode.

### 3.1. Latching Switches

When the acceleration increases to the threshold of the inertia switch, the mass will move and pass through the hook. When the acceleration is slowly reduced to 0, the latching mechanism prevents the proof mass from returning to the initial state. Then the switch stays in an “on” state. Lee et al. [80] introduced a hooked latch to an electrode for the contact part, which improved the reliability of the contact. The power consumption directly affected the practicability and reliability of the sensor system. An analog front-end is required to detect and interpret the output in the present commercial application of accelerometers. It leads the power consumption to be in the range of hundreds of µW to a few mW. Reddy et al. [81] developed a MEMS passive impact sensor that could be applied to multiple thresholds and had a good latch device. It could measure an impact in the range of 20–250 g at a threshold 10 times higher than itself. Ramanathan et al. [82] designed a new MEMS inertial switch design framework. The mass block was mainly supported by a group of four folded beams and it could design the accelerometer according to the parameters input by people. It was verified that the optimal threshold acceleration of the framework was about 60 g. Guo et al. [22] presented a latching switch with an easy-latching/difficult-releasing (ELDR) latching mechanism and cylindrical contacts. The ELDR locking mechanism could reliably lock the switch. The measured latching shock was over 4600 g and the response time was less than 0.2 ms. Currano et al. [83] discussed a MEMS inertial switch that was used to monitor the shock event over a specified threshold level with a latching mechanism. However, introducing a latch structure led a recovery issue, which is not beneficial to early testing and unlocking. Therefore, Zhang et al. [84] designed a novel heterogeneous integrated inertial microswitch with an adjustable acceleration threshold that could get the switch to keep a stable “on” state due to the pull-in phenomenon of a predefined bias voltage. Besides, the switch was easily unlocked by removing the bias voltage.

### 3.2. Bistable Inertial Switches

Additionally, the concept of a bistable mechanism has been brought to micro-inertial switches [85,86], which utilizes a bistable post-buckling structure to achieve a similar latching function. Taking advantage of the ability of the bistable mechanism to maintain two bistable positions without consuming power, the bistable structure has great superiority in designing an inertial switch with a self-locking function. Sandia National Laboratories presented a fully compliant bistable microswitch with a tapered beam in [87] and a three-segment bistable inertial switch [23] consisting of a large mass suspended on the four bistable beams, which could achieve the on-off function of a high g-value acceleration. Bistable microstructures as nonlinear elastic spring elements to precisely define two static stable states have been involved [25]. However, bistable structures have not been used to decrease the contact resistance. To reduce the contact resistance, a novel magnetic actuated bistable acceleration switch was proposed in [26], which included an asymmetric bistable mechanism with a parallel beam and three magnets arranged three-dimensionally. Zhao et al. [88] presented a wireless inertial microswitch incorporating a bistable flexible mechanism. When the bistable structure underwent buckling deformation under an excitation of external acceleration, the proof mass at the contact position could be stabilized. Liu et al. [89] studied a novel low-g MEMS bistable inertial switch with dual self-locking functions under an 8 g acceleration and reverse unlocking under a 105 g acceleration threshold, which was expected to carry out the reuse of the switch. The schematic of the designed bistable inertial switch is shown in Figure 9. This switch used a fully mechanical structure to greatly improve the anti-electromagnetic interference ability.

### 3.3. Liquid Inertial Switches

Unlike the conventional solid-to-solid-type micro inertial switch, a liquid inertial switch realizes contact through liquid–solid mode so as to improve its stability. The liquid–solid contact-based liquid switch mechanism can avoid problems such as signal bounce and contact wear during switch movement. Moreover, a liquid–metal (LM) microswitch [90] has been used in safety and arming (S&A) applications for time-delay fuses.

Yoo et al. [91] designed an inertial switch with a liquid–metal contact method, which greatly improved the stability of the switch contact. Kuo et al. [28] designed a passive inertial switch and integrated an *L*-*C* resonator to realize wireless signal transmission. The working fluid was water. In another design [92] introducing multiwall carbon nanotube (MWCNT)-hydrogel composite material, they proposed a liquid-type inertial switch integrated with a passive inductor/capacitor (*L*-*C*) resonator. This switch could achieve various threshold levels of acceleration by various channel configurations. Huang et al. [27] designed a microfluidic time-delay switch. The device consisted of glycerol (as the working fluid), a microcapillary valve (as the time-delay mechanism), and sensing electrodes. The schematics are shown in Figure 10. By changing the geometric design of the microfluidic system (such as channels and valves), the delay time of the microfluidic switch could be easily adjusted. The experimental test showed that the delay time range of the switch was 4.1 to 10.9 s.

Nie et al. [29] developed an innovative microfluidic inertial switch structure. It had the characteristics of accurate delay response. The results showed that the actual delay time was 114.1 ms under an inertia load of 75 g. Most studies used mercury as a conductive metal liquid, but it is toxic. Gallium indium (EGaIn) was used as the conductive element of the mobile electrode, which had the obvious advantages of non-toxicity, low viscosity, and high conductivity. Shen et al. [30] designed a self-recovery inertial switch, and its working liquid was a gallium-indium (EGaIn) metal droplet. The switch employed a U-type microchannel in order to reduce air resistance in the microchannel. There were sensing electrodes in the rectangular microchannel. After the acceleration signal disappeared, the droplet returned to the reservoir. This demonstrated the automatic-recovery characteristic of the switch. Liu et al. [93] used Galinstan and designed a microfluidic inertial switch. The test results showed that the response time was 0.66 ms and the contact time was 0.88 ms under a 51.2 g acceleration threshold. Liu et al. [31,94,95] conducted further research on the LM switch. An increase in the contact angle would increase the viscous force of the droplet on the capillary valve, thereby prolonging the response time. Furthermore, the switch structure introduced a two-stage microvalve, which facilitated adjustment of the threshold by changing the mercury volume. Huh et al. [96] designed a dual-axis accelerometer based on a liquid–metal droplet. The movement of droplet was leaded by the induced inertial force. Then they proposed an opposite design [97], which was to etch the cone-shaped guiding channel on the upper cover. The droplet was located in the channel. The switch could sense the acceleration signal in the range of 0–1 g, with high resolution and long service life.

The droplet used in the switch was selected according to its density, toxicity, melting point, and other characteristics. The following Table 5 lists the characteristics of several commonly used droplets in microfluidic switches. However, because liquid metals are volatile, the sealing problem of the switch requires special attention. Fluid is greatly affected by temperature, and cannot be kept constant at different temperatures. Therefore, the applications of fluid switches are seriously limited. A summary of some examples of typical persistent inertial switch performance is shown in Table 6.

## 4. Typical Fabrication Methods

The design and fabrication method can directly affect the reliability of MEMS inertial switches, especially the lifetime, robustness, and stability under extreme conditions of shock temperature, humidity, chemical exposure, or other challenges. Depending on the design and materials of the inertial switch, fabrication methods can be divided into three categories: (1) standard silicon micromachining technology, (2) non-silicon surface micromachining technology, and (3) the special fabrication method for liquid inertial switches.

### 4.1. Standard Silicon Micromachining Technology

Standard silicon micromachining is mainly manifested as a combination of bonding and deep eclipsing technologies to pursue large mass, stress reduction, and three-dimensional processing. Direct silicon bonding is completed by heating to promote the polymerization of hydrogen bonds to generate SiO_2_, so the surface treatment of the silicon wafer before bonding is key to making the surface of the silicon wafer absorb more OH^−^. Besides, gold–gold thermocompression bonding has been used in the preparation of bistable inertial switches to obtain a mass block of large thickness [89].

The main process of the standard silicon micromachine is sketched as follows and illustrated in Figure 11.
(1)SOI wafer preparation: A BOE rinse is performed to remove the native oxide layer on the SOI wafer.(2)Thermal oxidation: An SiO_2_ layer is grown and patterned, which serves as the etching mask layer for deep reactive ion etching (DRIE).(3)Oxidation pattern: A thin layer of Al_2_O_3_ is deposited on the SOI device layer via atomic layer deposition. Then the Al_2_O_3_ film is patterned as a hard mask for a silicon etch.(4)Device layer etching: The inertial switch silicon skeleton is then formed in the SOI device layer via silicon dry etching.(5)Backside lithography: Backside etching is carried out, followed by long DRIE technology, to remove the handle layer underneath the device and avoid any potential stiction issues for the large proof mass. Other Bosch technology is acceptable to ensure the verticality of the etching.(6)Moveable electrode: The moveable electrode (Ti/Pt/Au) is deposited on the back side.(7)Back side of the proof mass: Another DRIE process is applied to reveal the back side of the proof mass.(8)Microswitch release: The MEMS switch is finally released after removing the excessive SiO_2_ layer in the BHF solution.

**Figure 11 micromachines-13-00359-f011:**
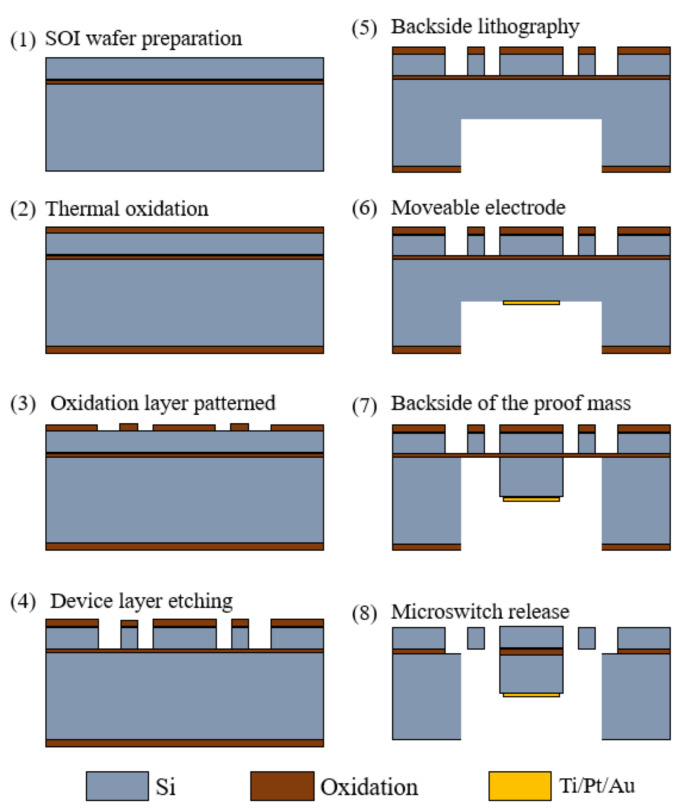
Schematic view showing the fabrication process of a standard silicon micromachine.

### 4.2. Non-Silicon Surface Micromachining Technology

Non-silicon surface micromachining technology on a different substrate such as silicon, metal, and glass, etc., mainly includes sputtering, electroforming, and sacrificial layer technologies. Some typical new inertial microswitches, including vertical [1] and lateral actuation [3], uniaxial [35], multiaxial [4], and omnidirectional sensitive [19] and flexible electrodes using CNT/Cu composites to enhance contact [78], are fabricated by this technology.

The following is a summary of the surface micromachining process of a MEMS inertial switch combined with the preparation of a uniform omnidirectional sensitivity inertial switch. The main procedure is sketched in Figure 12 and described as follows:(1)Preprocessing of the substrate: The roughness of the substrate surface is reduced via polishing techniques and by cleaning.(2)Photoresist lithography: A spin-coating photoresist on the substrate and photolithography are carried out. Table 7 shows some common photoresists, including their performance and coating thickness. Mostly, negative photoresist (SU-8) and positive photoresist [6] are used for the mold and sacrificial layers, respectively.(3)Micro electroforming: As the structure material, electroplated metal nickel (Ni) has good mechanical properties and can effectively solve the problem of switch breakage under a high acceleration impact. Volume error can be reduced by controlling the plating time.(4)Seed layer: Sputtered Cr/Cu on the substrate is used as a seed layer for device electroplating.(5)Multilayer repetition of micro electroforming: Multilayer plating technology can overcome etching difficulties of a high slim ratio of inertial switches.(6)Microswitch release: The photoresist and seed layer are removed, and then the inertial microswitch can be obtained. Usually, acetone or boiled inorganic are used to remove negative photoresist SU-8 and an ammonia/peroxide solution is used to remove the seed layer.(7)Rinsing and drying the device: The released microswitch is rinsed with isopropyl alcohol or deionized water, and then dried to avoid stiction.

**Table 7 micromachines-13-00359-t007:** The performance and coating thickness of common photoresists.

Type	Name	Performance	Coating Thickness
Positive	AZ P4620	Ultra-thick film, high-contrast, and high-speed positive-tone standard photoresist for semiconductor and/or GMR head manufacturing processes.	10–15 μm
	AZ 50XT	Stable, excellent coating characteristics and sidewall profiles for developing plating and wafer-bumping applications.	40–80 μm
	AZ 9260	Small absorption coefficient and a typical photoresist for thick resist etching processes.	6.2–15 μm
Negative	SU-8 series	High aspect ratio imaging, improved adhesion, reduced coating stress, vertical sidewalls, and faster drying for increased throughput.	0.5–300 μm
Variable	AZ 5214E	Wide viscosity variation suitable for high resolution process (lift-off process) and available for positive/negative patterning.	0.5–6 μm

**Figure 12 micromachines-13-00359-f012:**
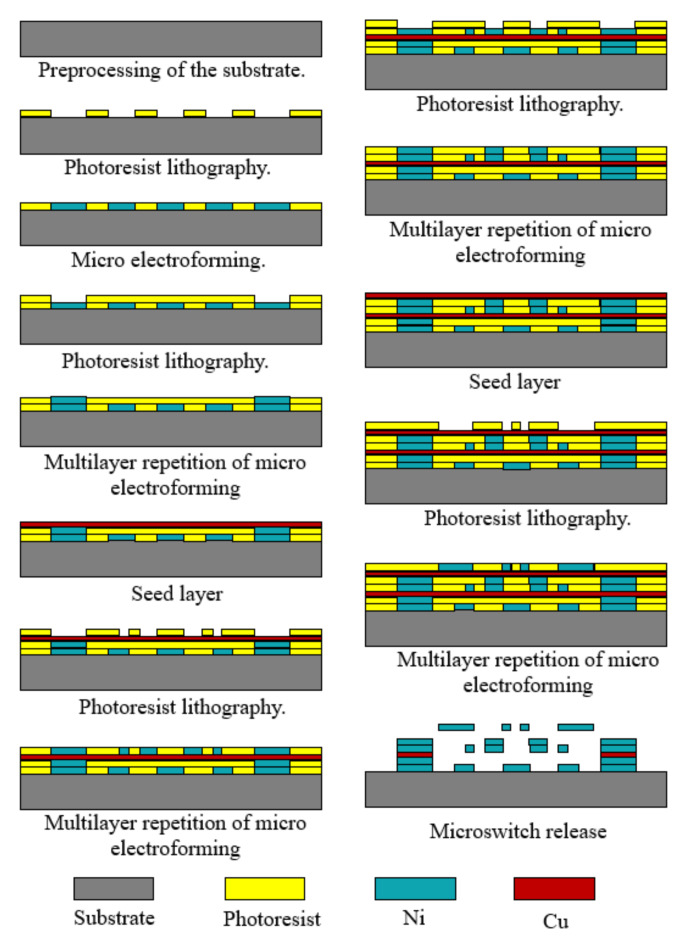
Sketch of the main fabrication process of surface micromachining.

### 4.3. Special Fabrication Method for Liquid Inertial Switches

For liquid inertial microswitches, the key difficulty in the fabrication process is the problem of the tightness of the liquid electrode (movable electrode). Micro-channels are formed on a silicon substrate by microfabrication technology, and the covered glass substrate is convenient for observing the movement of droplets. The specific fabrication process is illustrated in Figure 13 and expressed as follows.

Microchannels with different cross-sections can be etched by micromachining processes. The silicon wafer is etched by DRIE technology, and the bonding between the silicon wafer and the glass cover is realized by anodic bonding.
(1)Silicon substrate photolithography: The microchannel is SOI material, with a photoresist masking pattern for ICP etching microchannels on the SOI material.(2)Silicon substrate ICP etching: The microchannel is etched by ICP technique, and then the photoresist is removed.(3)First metal electrode layer: The photolithography, sputtering, and lift-off techniques are applied on the glass substrate. The first metal electrode layer is achieved on the glass.(4)Second metal layer: The same technology as above is used to achieve the second metal layer on the glass substrate.(5)Glass cover plate laser drilling: To achieve the adjustment of the volume of the flowing droplets, the adjustment holes and channels are laser etched.(6)Anode bonding and dicing: After wafer-level packaging, chips are obtained through precise dicing technology.

After completing the structural preparation of the fluid switch, the droplets are injected into the adjustment hole and the channel is adjusted to make the droplet enter the initial position. Although gold has good conductivity, its application is limited due to its high solubility with mercury. Therefore, chromium (Cr) can be chosen as the electrode material due to its proper electrical conductivity and low solubility with mercury. In addition, the preparation process of an alternative fluid inertial switch including a package shell is given, as shown in Figure 14 [27]. The detailed process is as follows, which mainly consists of three parts, including a glass wafer, silicon wafer, and package.
(a)Glass wafer(1)Adhesive layer and electrodes: A chrome film as an adhesion layer and a gold film as a sensing electrode are evaporated and patterned on a glass substrate by lift-off technology.(2)Parylene film: Parylene thin films are deposited by chemical vapor deposition (CVD) and patterning by oxygen plasma. In the subsequent silicon-to-glass bonding process, the parylene film serves as a hydrophobic surface and bonding interface.(b)Silicon wafer(3)Thermal oxidation: Silicon dioxide is processed by thermal growth and then patterned on a silicon wafer.(4)Microchannel and fluidic components: The DRIE process is used to prepare structures required for microfluidic work, such as capillary valves, reservoirs, and vents.(5)Parylene film: Parylene film plays a role in surface modification and bonded adhesion layers, which is deposited by the CVD process.(c)Packaging(6)The Si substrate is filled with fluid and then sealed to glass by bonding technology.(7)The liquid inertial switch device is created after dicing the Si substrate and electrical routing.

**Figure 13 micromachines-13-00359-f013:**
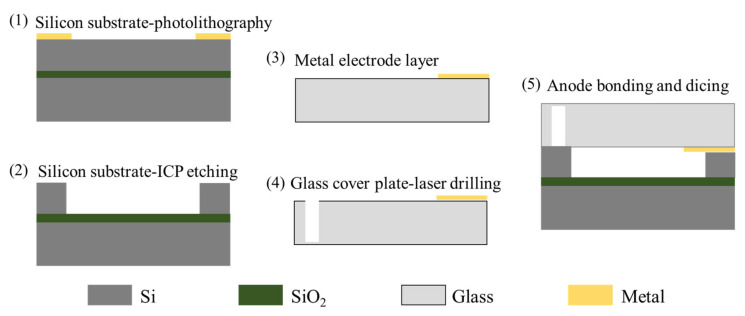
The special fabrication method for a liquid inertial switch.

**Figure 14 micromachines-13-00359-f014:**
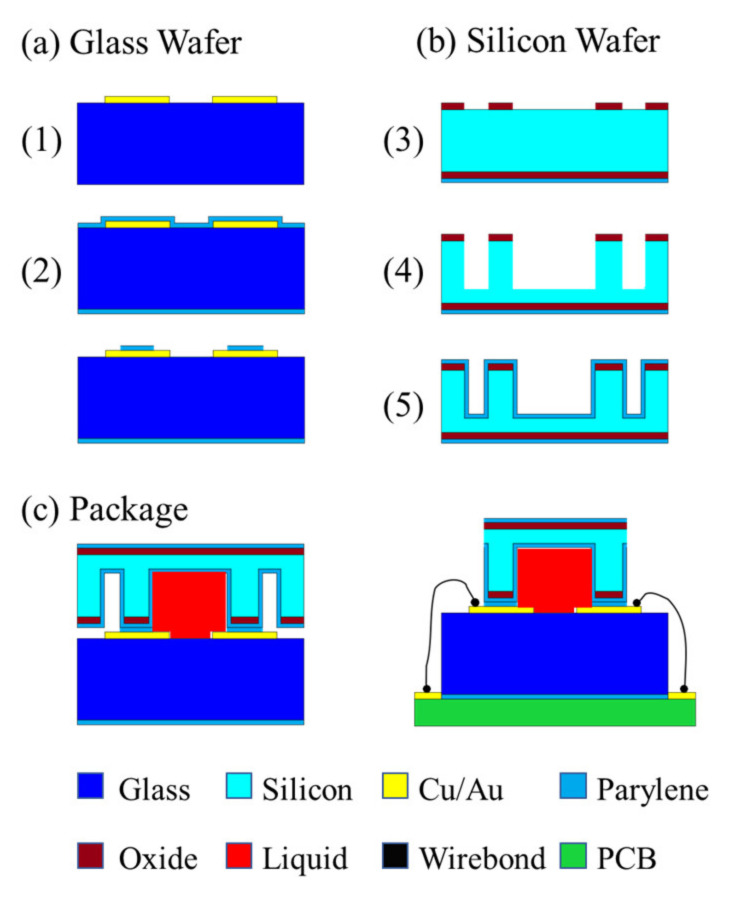
Manufacturing and packaging process for a fluid switch.

MEMS manufacturing process technology is the basis for supporting the development of mechanically robust, high-performance inertial switch designs. It is combined with the hermetic silicon process, which is the key to realizing a truly reliable long-life MEMS switch. An airtight protective casing is formed around the MEMS inertial switch device. Regardless of the external packaging technology used, this airtight enclosure can improve the environmental robustness and service life of the inertial switch.

## 5. Challenges and Prospects

In engineering practice, it is always difficult to break through bottlenecks such as inertial switch manufacturability, threshold repeatability, and threshold inconsistency. Both the bottlenecks and the opportunities for MEMS inertial switches are in progress. In order to successfully commercialize MEMS inertial switches, there are some challenges that remain and prospective trends that are worthy of attention.
(1)Persistent switches can achieve stable closure. However, they are not suitable for an engineering environment wherein repeated application is required. In addition, in early experimental tests, its one-time use feature will cause a great waste of devices, so it is necessary to study the self-unlocking method for this type of switch.(2)DRIE and Bosch technology are able to achieve large mass preparation, and multi-layer electroforming technology makes complicated inertial switches possible with multiple direction and threshold sensitivities. However, the inherent fabrication errors and the residual stress lead to low threshold accuracy. Thence, the method of error compensation can significantly improve the threshold accuracy and sensitivity.(3)Mature SOI technology can effectively guarantee the fabrication accuracy of the inertial switch. However, the packaging process of the inertial switch in the later stage affects the final size and application of the switch. Usually, the package shell of the inertial switch needs to have high sealing and pressure resistance to ensure air film damping, and the packaged chip is also easier to install and transport. However, most of the current switch research is limited to experimental test packaging, which is far from the packaging requirements of the actual application environment. Therefore, it is of great significance to select appropriate materials and packaging processes to bring the switch from research and development to practical applications.(4)The surface micromachining process is compatible with the integrated circuit production process, and the integration is high. Furthermore, integrating and applying the inertial switches with integrated circuits for systematic integration is a significant trend as well.

## Figures and Tables

**Figure 1 micromachines-13-00359-f001:**
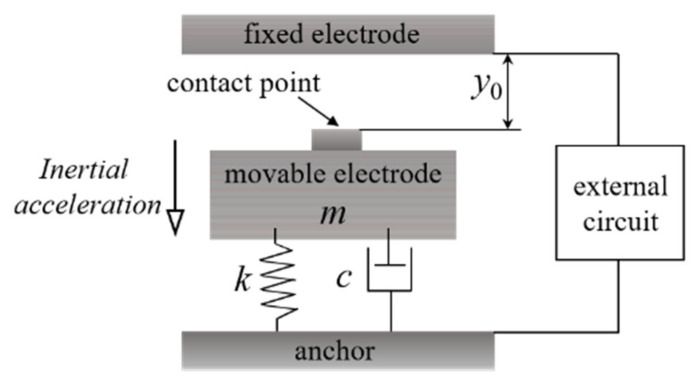
Schematic diagram of the spring (*k*)–mass (movable electrode *m*)-damping (*c*) system model of a uniaxial inertial switch. *y*_0_ means the initial distance between the movable electrode and the fixed electrode.

**Figure 3 micromachines-13-00359-f003:**
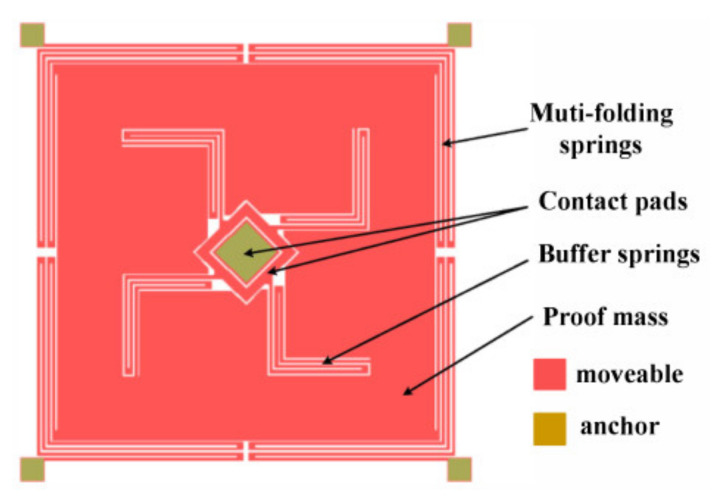
Schematic of the two-axis low-g inertial switch.

**Figure 4 micromachines-13-00359-f004:**
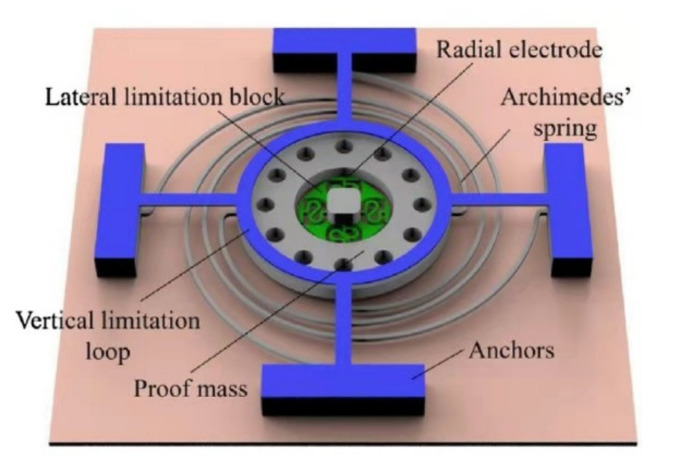
Structural sketch of the omnidirectional inertial switch.

**Figure 5 micromachines-13-00359-f005:**
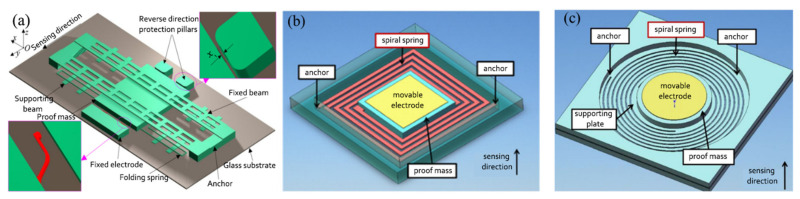
Schematic diagrams of three low-g inertial switches. (**a**) The nickel low-g MEMS inertial switch [57]. (**b**) The silicon-based low-g MEMS inertial switch [58]. (**c**) The lower stiffness of the redesigned inertial switch [59].

**Figure 6 micromachines-13-00359-f006:**
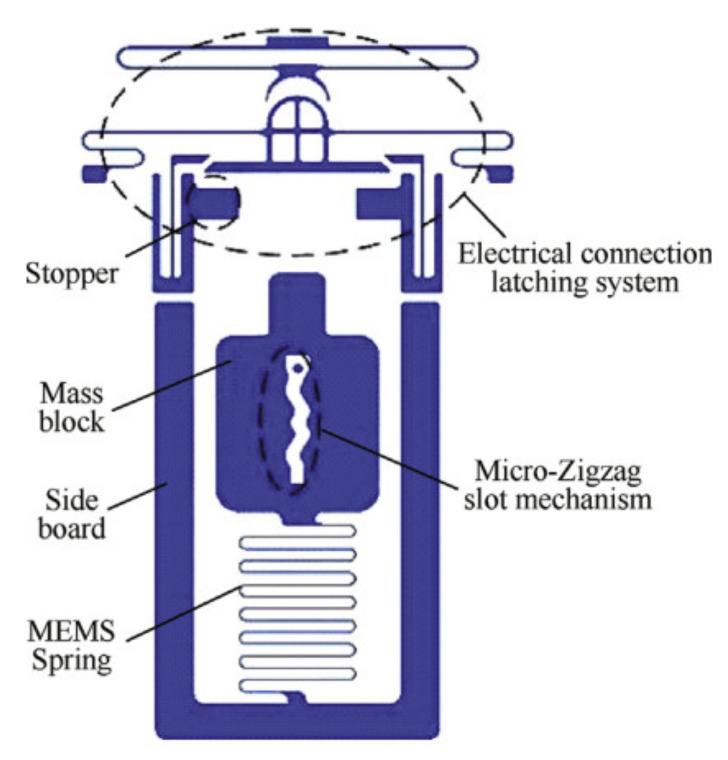
Schematic diagrams of the MEMS inertial high-g switch.

**Figure 7 micromachines-13-00359-f007:**
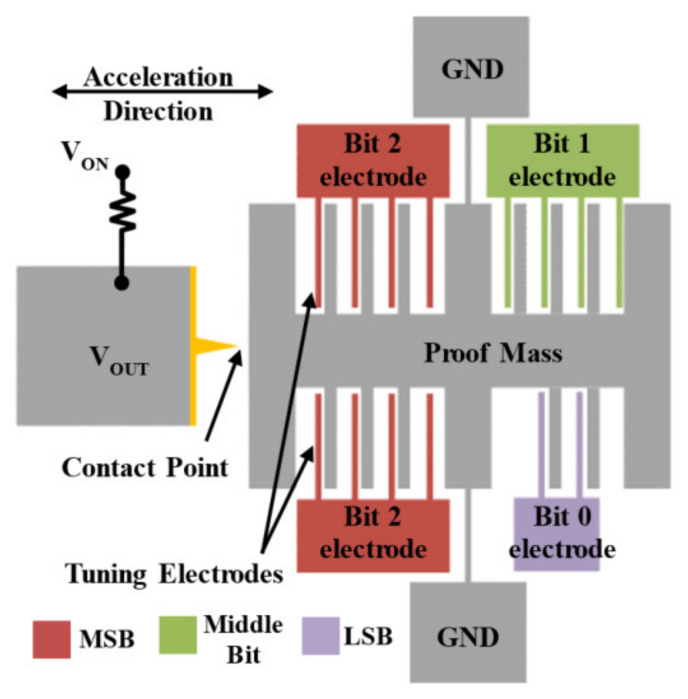
Simplified schematic view of a 3-bit digitally operated MEMS switch.

**Figure 8 micromachines-13-00359-f008:**
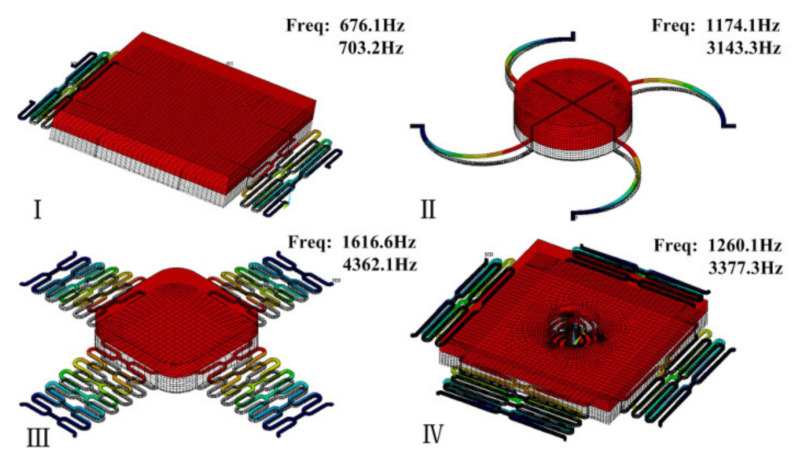
Modal analysis of various microswitches and their first two frequencies. Type I adopted a vertically driven structure with low off-axis sensitivity. Type II and III used a fixed contact point on the proof mass. Type IV realized a longer duration contact with a movable contact point.

**Figure 9 micromachines-13-00359-f009:**
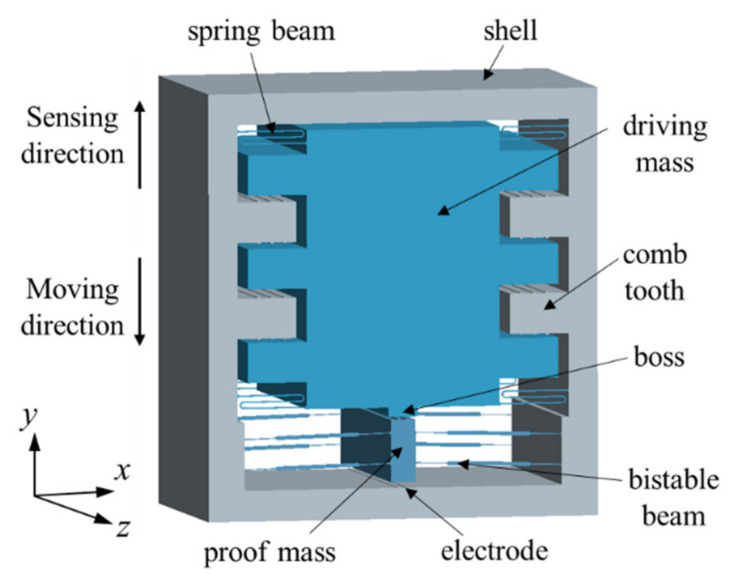
Schematic and structural parameters of the designed MEMS bistable inertial switch.

**Figure 10 micromachines-13-00359-f010:**
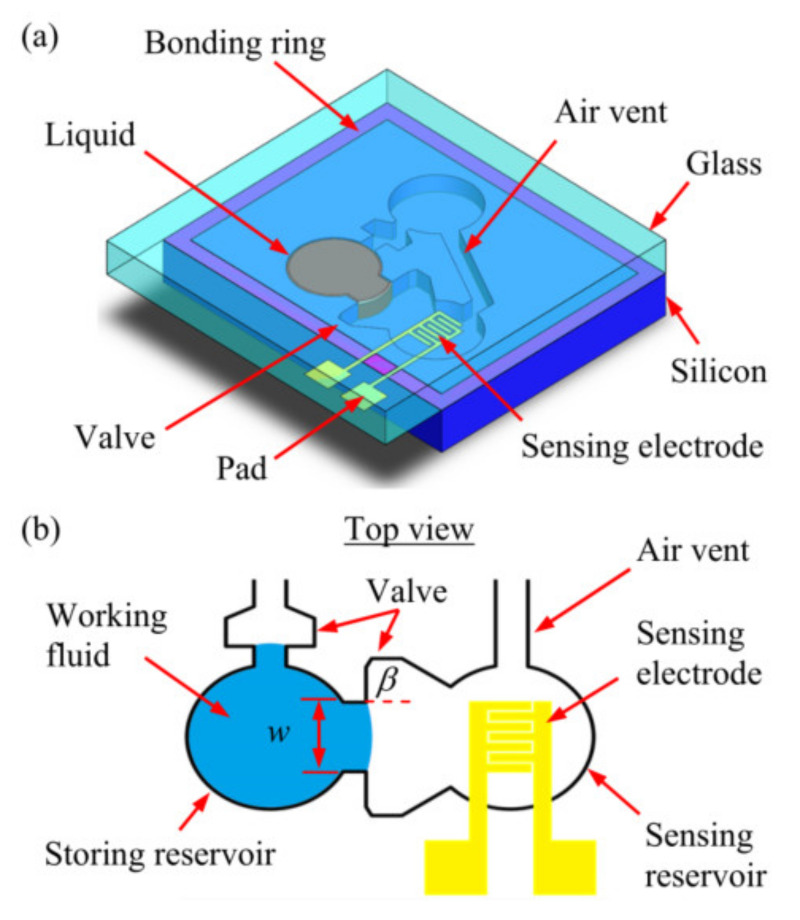
Schematics of the proposed microfluidic system design: (**a**) the S-A device and (**b**) top view of the device.

**Table 1 micromachines-13-00359-t001:** A summary of some examples of typical sensitive direction inertial switch performance.

Sensitive Direction	References	Material	Acceleration Threshold	Contact Time	Special Design and Function	Application
Uniaxial	Wang et al. [5]	Ni	180 g	1050 μs	Compliant cantilever fixed electrode to contact enhancement	—
	Wang et al. [7]	Ni	38 g	230 μs	Elastic fixed electrode	Safety airbags
	Kim et al. [12]	Si	2.0–17.25 g	—	—	Environments and applications require accurate threshold
	Yang et al. [32]	Ni	100 g	12 μs	Bridge-type elastic beams to enhance contact time	—
	Cai et al. [34]	Ni	70 g	30 μs	Stationary electrode changed from two bridge-type beams to one cross beam to reduce the off-axis sensitivity.	—
	Zhang et al. [35]	Ni	165 g	35 μs	Double-stair shape cantilever beam	Internet of Things (IoT) system to remote detection of vibration shock
	Yang et al. [36]	Ni	272 g	20 μs	Double-layer suspended springs for improving single-axis sensitivity	IoT system
	Fathalilou et al. [37]	—	154 g	—	A dual-mass switch with auxiliary mass spring	Automobile, medicine, and aerospace
	Ren et al. [38]	Ni	40 g	80 μs	Self-powered	Vibration energy harvester (VEH) and potential wake-up application
	Raghunathan et al. [39]	SOI	60–131 g	—	Surviving acceleration loads 200 times greater than its designed trigger load	Ballistic rockets
	Chen et al. [42]	Ni	297 g	80 μs	Compliant cantilever beam	Automotive safety crash airbags
Biaxial	Lin et al. [48]	Si	60 g	—	Buffering springs to extend the contact time	—
	Niyazi et al. [49]	Si	69 g and 121 g	—	Separate digital outputs for each threshold	Active suspension systems
	Xu et al. [50]	Si	800–2600 g	—	High-resolution digital quantitative acceleration measurements	IoT system
Triaxial	Chen et al. [2]	Ni	255–260 g (+*x* and +*y* axis) ~75 g (+*z* axis)	~60 μs (+*x* and +*y* axis) ~80 μs (+*z* axis)	Flexible fixed electrode can prolong the contact time and eliminate the rebound	IoT system
	Currano et al. [51]	Si	50–250 g	255 μs	Compliance in all axes identical	Early warning for traumatic brain injury (TBI)
Omnidirectional	Xi et al [19]	Ni	450 g	60 μs	A dual mass–spring system	—
	Liu et al. [52]	Si	20 g	—	The response time of 0.46 ms is short enough	—
	Du et al. [54]	Ni	35–40 g	~100 μs	Electrode with a spherical contact surface	Automotive airbags
	Du et al. [55]	Ni	7.9–11.3 g	>300 μs (XOY plane) >230 μs (axial)	Method of “thickness compensation” to control threshold accuracy	Wearable systems and airbags
	Chen et al. [56]	Ni	58 g *x* direction 37 g *z* direction	18 μs	Rectangular spring to reinforce switching system’s stability	Transport of special goods and drop detection
Multidirectional	Yang et al. [53]	Ni	70 g	110 μs	Polymer–metal composite fixed electrode	—

**Table 2 micromachines-13-00359-t002:** A summary of some examples of typical threshold acceleration inertial switch performance.

Threshold Acceleration	References	Material	Acceleration Threshold	Special Design and Function	Application
Low-g	Chen et al. [57]	Ni	18 g	L-shaped elastic cantilever beam fixed electrode	Health monitoring and special industrial transportation
	Xiong et al. [58]	Double buried SOI	7.4 g	Low-stiffness spiral spring	Linear acceleration sensing
	Zhang et al. [59]	Double buried SOI	5 g	Circular spiral springs	—
	Hwang et al. [62]	Si	6.61 g	Displacement-restricting structures for all directions to prevent breakage of the spring	Military applications
	Massad et al. [63]	Gold	6–10 g	Four folded beams as springs	RF MEMS
High-g	Nie et al. [64]	Ni	3000 g	Zigzag groove to distinguish the fuse launch acceleration and the accidental fall shock	Medium- and large-caliber projectile fuses
	Singh et al. [65]	SOI	3500 g	Independent angled latching mechanism	Critical applications without electricity
	Xu et al. [67]	Ni	500 g	Synchronous follow-up compliant electrodes for extending the contact	—
	Xi et al. [68]	Ni	1200 g	Detecting the acceleration threshold and direction	Directional warheads impacting targets at high speed
Threshold tuning	Kim et al. [12,69]	Si and glass	2–17.25 g	Comb drive actuators to tune the acceleration threshold	Secure/armed position convertibility for military applications
	Kumar et al. [70]	Si	0~1 g	Bias voltage and working voltage are used to adjust acceleration	Integrated systems
	Ma et al. [71]	Si	40–75 g	MEMS digital-to-analog converter (M-DAC) to adjust acceleration thresholds	Crash recorders and arming and firing systems

**Table 3 micromachines-13-00359-t003:** Comparison of the contact time of inertial microswitches from different designs.

Inertial Microswitches of Different Designs	Simulated Contact Time (μs)	Measured Contact Time (μs)	Sketch of the MicroswitchesMovable Electrode  Fixed Electrode 
Conventional microswitch	~1	--	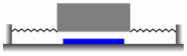
Microswitch with a bridge-type compliant fixed electrode	~5(t_1_), ~10(Δt),~2(t_2_) skip contact	~13(t_1_), ~60 (Δt),~8(t_2_) skip contact	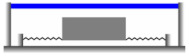
Improved microswitch with cantilevers	~160 noskip contact	~240 noskip contact	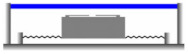

**Table 4 micromachines-13-00359-t004:** A summary of some examples of typical contact-enhanced inertial switch performance.

Methods of Contact Enhancement		References	Material	Acceleration Threshold	Contact Time	Application
Special structure	Double mass–spring system	Cai et al. [1]	Ni	145 g	>50 μs	Automotive airbag system
	L-shaped flexible cantilever fixed electrode	Wang et al. [5]	Ni	180 g	1050 μs	Circuit analyzing in many applications
	L-shaped compliant cantilever beam	Chen et al. [3]	Ni	259 g	75 μs	Small-scale or longlifetime systems
	Bridge-type elastic fixed electrodes	Yang et al. [32]	Ni	100 g	12 μs	Accessories and automobile applications
	Cantilever beams on the mass block as the buffer	Yang et al. [73]	Ni	55 g	240 μs	—
	Two L-shaped elastic cantilever beam electrodes	Xu et al. [6]	Ni	288 g	150 μs	Small-scale or long-lifetime systems with limited supply power
Materials	Carbon nanotubes (CNTs)	Lee et al. [75]	SOI	—	108 μs	Airbag restraint systems and geriatric healthcare systems
	Polymer metal composite	Yang et al. [77]	Ni	70 g	110 μs	Detecting multidirectional vibration shocks
	Carbon nanotubes/copper(CNTs/Cu)	Wang et al. [78]	Ni	80 g	112 μs	—
Electrostatic force assistance		Li et al. [79]	Ni	22 g	540 μs	Hard conditions or remote monitoring

**Table 5 micromachines-13-00359-t005:** Characteristics of some droplets in microfluidic switches.

Working Droplet	Density (g/cm^3^)	Melting Point (°C)	Toxicity	Surface Tension (mN/m)	Others Characteristic
Water	1.0	0	Non-toxic	73	--
Mercury	5.43	−38.83	Toxic	485.5	Opaque High reliability
Glycerol	1.26	−17.8	Non-toxic	63.4	High dielectric constant
gallium-indium (EGaIn)	6.25	15.7	Non-toxic	445	Low viscosity High conductivity
Galinstan [98]	6.44	−19	Non-toxic	534.6	Low vapor pressure Easy to oxidize

**Table 6 micromachines-13-00359-t006:** A summary of some examples of typical persistent inertial switch performance.

Type	References	Material	Acceleration Threshold	Performance	Application
Latching switches	Lee et al. [80]	Si and glass	43.7 g	Mechanical hooked latch	Airbags, parachutes, and military devices
	Reddy et al. [81]	SOI	20–250 g	Robust latching mechanism with mass-spring assembly	Long-term remote monitoring applications
	Ramanathan et al. [82]	SOI	60 g	Semi-circular latch key	Projectiles or the separation of rocket stages
	Guo et al. [22]	Si and glass	4600 g	Easy-latching/difficult-releasing (ELDR) latching mechanism	—
	Zhang et al. [84]	Si and Ni	57 g	Stable “on” state due to a predefined bias voltage	Monitoring the transportation of special goods
Bistable inertial switches	Zhao et al. [88]	Ni	32.38 g	V-shaped slender bistable beams	Remote detection of threshold acceleration and corresponding response time
	Liu et al. [89]	SOI	8 g (self-locking) 105 g (self-locking)	Three-segment fully compliant bistable beams	Military applications
Liquid inertial switches	Yoo et al. [91]	Si and glass	—	Liquid–metal (LM) droplet combined with selective surface modification inside the channel	—
	Kuo et al. [92]	Si, glass, and PDMS	∼60 g	Multiwall carbon nanotube (MWCNT)–hydrogel composite integrated with an inductor/capacitor (L–C) resonator	Sensing acceleration inducing by impact
	Nie et al. [29]	Si (EGaIn)	75.1 g, 46.6 g, 36.5 g	Precise time-delay response characteristic	Fuze safety and arming systems
	Liu et al. [93]	Glass and PDMS	51.2 g	Automatic-recovery inertial switch and Galinstan marbles	—

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
