# Peer review of "Research Progress of MEMS Inertial Switches"

_micromachines, 2022, doi:10.3390/mi13030359_

Round 1

Reviewer 1 Report

This work presents a review article that investigates inertial switches. Various structures and mechanisms are discussed an investigated in this article. In general, most of the references in this area have been covered. To complete the all of important topics in this area, beside the four subtitles (Intermittent inertial switches, Persistent inertial switch, Typical fabrication methods, Challenges and Prospects) the following subtitle (for example: performance range of inertial switches) should be added to the text, that the performance range of inertial switches is discussed and compared in this subtitle.

However in the present form there is essential information missing and also the edition needs to be improved.

Author Response

Dear Reviewer,

Thank you for your valuable comments, we have sincerely responded to each comment and made some modifications with attached submitted pdf file named "Response to Reviewer 1 comments". We hope the revised manuscript would satisfy you and get your approval.

Thank you again!

Best regards.

Reviewer 2 Report

My attached is the comments to authors.

Author Response

Dear Reviewer, 

Thanks  a lot for your valuable comments, we have sincerely answered  every comment and made some modifications with attached submitted pdf file named "Response to Reviewer 2 comments". We hope the revised manuscript would satisfy you and  get your approval.

Thank you again!

Best regards.

Reviewer 3 Report

In this paper, the authors present “Research progress of MEMS inertial switches”. The quality of this paper is generally satisfactory; however, in order to lead to an improvement for this paper, I have following comments and suggestions.

1.  The author should include performance comparison table for MEMS initial switches. The author should also include a table summarizing their failure factors and improvement methods in order to better understand for a reader.

2.  A parallel paper titled “Recent Advancements in Inertial Micro-Switches” was published by author Peng.et.al. Challenges and Prospects section is also almost similar. What the author brings new things in this manuscript? Author should talk more about the challenges and prospects. 

Author Response

Dear Reviewer, 

Thanks  a lot for your valuable comments, we have sincerely answered  every comment and made some modifications with attached submitted pdf file named "Response to Reviewer 3 comments". We hope the revised manuscript would satisfy you and  get your approval.

Thank you again!

Best regards.

Round 2

Reviewer 2 Report

My attached is the comments.

Author Response

Dear Reviewer,

Thank you for your valuable comments. We have responded to your comments one by one in the"Response to Reviewer 2 Comments (2)". We hope our answers can be recognized by you.

We hope the revised manuscript would satisfy you and the reviewers.

Thank you!

Best regards,

Weidong Wang

Full Professor of the School of Mechano-Electronic Engineering

Xidian University

Reviewer 3 Report

Author has well responded to all my answers.

Author Response

Dear Reviewer,

Thank you for your effective suggestions before, and thank you even more for your recognition of our reply.

Best regards,

Weidong Wang

Full Professor of the School of Mechano-Electronic Engineering

Xidian University
